# The Synergistic Effects of Curcumin and Chemotherapeutic Drugs in Inhibiting Metastatic, Invasive and Proliferative Pathways

**DOI:** 10.3390/plants11162137

**Published:** 2022-08-17

**Authors:** Maria Younes, Rita Mardirossian, Liza Rizk, Tia Fazlian, Jean Paul Khairallah, Christopher Sleiman, Hassan Y. Naim, Sandra Rizk

**Affiliations:** 1Department of Natural Sciences, School of Arts and Sciences, Lebanese American University, Byblos P.O. Box 36, Lebanon; 2Department of Biochemistry, University of Veterinary Medicine Hannover, 30559 Hannover, Germany

**Keywords:** curcumin, combination therapy, cancer, metastasis, invasion, phytotherapy

## Abstract

Curcumin, the main phytochemical identified from the *Curcuma longa* L. family, is one of the spices used in alternative medicine worldwide. It has exhibited a broad range of pharmacological activities as well as promising effects in the treatment of multiple cancer types. Moreover, it has enhanced the activity of other chemotherapeutic drugs and radiotherapy by promoting synergistic effects in the regulation of various cancerous pathways. Despite all the literature addressing the molecular mechanism of curcumin on various cancers, no review has specifically addressed the molecular mechanism underlying the effect of curcumin in combination with therapeutic drugs on cancer metastasis. The current review assesses the synergistic effects of curcumin with multiple drugs and light radiation, from a molecular perspective, in the inhibition of metastasis, invasion and proliferation. A systemic review of articles published during the past five years was performed using MEDLINE/PubMed and Scopus. The assessment of these articles evidenced that the combination therapy with various drugs, including doxorubicin, 5-fluorouracil, paclitaxel, berberine, docetaxel, metformin, gemcitabine and light radiation therapy on various types of cancer, is capable of ameliorating different metastatic pathways that are presented and evaluated. However, due to the heterogeneity of pathways and proteins in different cell lines, more research is needed to confirm the root causes of these pathways.

## 1. Introduction

Curcumin, extracted from the rhizome of *Curcuma longa* L., has been traditionally used as an alternative medicine for the treatment of different diseases. Curcumin has, beyond this traditional use, various roles and functions. In fact, it is a strong anti-inflammatory, anti-oxidant and anti-proliferative agent [1,2,3]. Previous studies show that curcumin also exerts cytotoxicity on a broad range of cancer cell lines, such as lung, colon, breast and ovarian cancer [4,5,6,7]. However, despite its various functions, it has limited applications due to its low solubility in water, thus leading to low absorption and low oral bioavailability. Several experiments have been performed to improve the solubility of curcumin and enhance its therapeutic effects in different pathologies [8].

Curcumin targets several signaling pathways involved in the regulation of cell proliferation, invasion, metastasis and apoptosis via the regulation of different regulators, as detailed in Figure 1. Several studies investigated the antitumor activity of curcumin on different cancer cells, revealing its mechanism of action on various signaling pathways. Recent reviews have summarized the dysregulation of the SIRT, JAK/STAT, MAPK, P13K/Akt, Wnt/β-catenin, Notch and NF-κB pathways, which are highly involved in the regulation of cell proliferation, apoptosis, cell cycle arrest, oxidative stress as well as cell invasion [9,10,11,12]. Furthermore, the anti-proliferative, anti-metastatic and anti-invasive properties of curcumin are due to the inhibition of activator protein-1 (AP-1) and focal adhesion kinase (FAK) activity as well as the downregulation of inflammatory cytokines, such as CXCL1 and CXCL2, and interleukins- (ILs) 6 and IL-8 in various cancer cells [13].

An important signaling pathway that curcumin targets is the NF-κB signaling pathway (Figure 1). Nuclear factor kappa-light chain activation suppresses apoptosis and induces cell invasion, metastasis and cell proliferation. In fact, activated NF-κB promotes the constitutive activation of IκB kinase, which causes the phosphorylation and degradation of IκBα (inhibitor of κB, α). Several studies on breast and prostate cancer cell lines show that curcumin inhibits the stimulation of the upstream regulator of NF-κB, thus reducing its signal and downregulating the expression of IκBα kinase, leading to cell apoptosis [14]. Moreover, curcumin downregulates the expression of NF-κB–regulated gene products, including IκBα, Bcl-2, Bcl-xL, cyclin D1, matrix metalloproteinases (MMP) -2 and -9 and urokinase-type plasminogen activator (uPA), in addition to interleukin (IL)-6 and IL-8 [15,16,17].

Important gene-products of the NF-κB pathway involved in the regulation of tumor invasion and metastasis are uPA, MMP-2 and MMP-9. The uPA kinase binds to the uPA receptor and activates the protease plasmin, which degrades the extracellular matrix (ECM). Similarly, MMPs are endopeptidases that degrade ECM, hence promoting tumor cell invasion and metastasis [18]. In this context, previous studies conducted on colon cancer cells revealed the potential role of curcumin in suppressing metastasis via the AMPK activation and subsequent inhibition of NF-κB, uPA and MMP-9 [17]. Similarly, the anti-invasive properties of curcumin were demonstrated in MCF-7 breast cancer cells via a dose-dependent decrease in uPA protein levels [19].

The signal transducer and activator of the transcription 3 (STAT3) signaling pathway is also involved in metastasis as well as migration and invasion into the ECM. Several cancer-derived cell lines depend on the constitutive activation of STAT3, which is overexpressed in tumor cells as a result of its phosphorylation by Janus kinases (JAKs). A previous study suggests that STAT3 regulates cell proliferation and the expression of several proteins, namely c-myc, cyclin D, Bcl-2, vascular endothelial growth factor (VEGF) as well as MMP-2 and MMP-9 [20,21]. It was shown that curcumin is able to suppress the expression of STAT3 in various cell lines, including pancreatic, human non-small cell lung cancer, myeloid leukemia and breast cancer cells (Figure 1) [22,23,24,25].

Nevertheless, the expression of BCAT1 in cancerous cells has a major role in the progression of myeloid leukemia cells. The therapeutic effects of curcumin were able to induce apoptosis in leukemia cells by the suppression of the BCAT1 and mTOR pathways (Figure 1). In fact, mTOR is capable of encouraging resistance of cells to different drugs, which would partake in the proliferation and migration of cancerous cells to different tissues. Studies have indicated that curcumin is able to control proliferation through the cleavage of PARP1, which is a nuclear enzyme responsible for DNA repair, and its upregulation has been reported in various human cancer cell lines [26].

Along with its extensive roles in regulating cancerous pathways in different cell lines and ameliorating the well-being of patients, curcumin serves an important role in enhancing the chemotherapeutic effects of other drugs, such as doxorubicin, cisplatin and paclitaxel, used in cancer therapy by increasing the sensitivity of cell lines to these prospective drugs, as previously detailed in a review published by Farghadani et al. [27]. The combination of curcumin with several drugs has reduced the toxic effects of the drugs themselves and has reduced resistance, making them more effective. In fact, previous studies reported that curcumin was able to enhance the sensitivity of cisplatin-resistant non-small cell lung carcinoma [28], while it was found effective in increasing the sensitivity of ovarian cancer cells to cisplatin when curcumin was combined with resveratrol [29]. The synergism between curcumin and many chemotherapeutic drugs has also enhanced pathways in cancerous cells, such as the induction of apoptosis and the suppression of proliferation, invasion and metastasis.

Metastasis is the process of the transformation of cells into a malignant form that involves a series of genetic and epigenetic modifications due to the increase in genomic instability. These reformations lead to irregular cell cycle control, difficulty in apoptosis and acquire infinite replication abilities. This also enables the cells to invade, migrate and spread their properties to other cells and tissues [30]. Unfortunately, drugs that target metastatic pathways have portrayed long-term side effects that have led to abysmal repercussions in the well-being of patients undergoing chemotherapy. For this reason, experts resorted to the combination of these chemotherapeutic drugs with curcumin due to curcumin’s ability to reverse the toxic effects of such drugs as well as its capability of enhancing the suppression of metastatic pathways, as described previously by Liu et al. [31]. The following review discusses the synergistic effects of curcumin combined with multiple drugs in the inhibition of various types of cancer cells’ proliferation, invasion and metastasis, highlighting the underlying molecular pathways.

## 2. Results

The following section reviews the literature from the past five years reporting the promising effects upon combination of curcumin with various chemotherapeutic drugs or radiation on metastatic, invasive and proliferative cancer therapy in multiple cancer types. The combination of curcumin with each of 5-fluorouracil, doxorubicin, paclitaxel, metformin, docetaxel, berberine, gemcitabine and light radiation is described in the following subsections, and the data are summarized as a Appendix A.

### 2.1. The Hindrance of Metastatic/Proliferative/Invasive Pathways Using Curcumin and Light Radiation

The usage of radiotherapy in cancer treatment has portrayed crucial properties against various cell lines and their respective metastatic pathways. Studies have shown that radiotherapy has become the standard treatment to achieve apoptotic and non-metastatic effects. However, these effects are not always guaranteed due to the heterogeneity of responses reported in different patients, in addition to the adverse side effects of radiation therapy on normal tissues. Increased evidence has shown that many patients develop a relapse, which would hinder the radiotherapeutic effect and lead to abysmal repercussions in the migration of the cancerous cells to other tissues [27]. For this reason, it was of great importance to resort to alternative therapies to avoid these drastic side-effects by decreasing the doses of radiations while remaining effective. The combination of curcumin with light radiation has elicited promising effects in controlling metastasis and invasion through the suppression of multiple pathways, as summarized in Figure 2. In fact, curcumin has ameliorated the efficacy of light radiation and has increased the sensitivity of the cancerous cells to respond to radiation.

Studies have shown that radiation therapy induces EMT in cancer cells by downregulating E-cadherin and upregulating mesenchymal molecular markers. E-cadherin has a serious role in cancer since it stabilizes cell–cell adhesion in epithelial cells in addition to suppressing tumor transformation and growth. Moreover, MMP9 plays a determinant role in cancer invasion as described above. Upon addition of curcumin to radiation therapy, the expressions of E-cadherin, vimentin and SLUG, which are crucial EMT markers and promoters of invasion, were decreased, ultimately leading to the inhibition of EMT properties in A549 lung cancer cells. Nevertheless, the synergistic effect of curcumin and radiation suppressed MMP9 protein levels, which, in turn, inhibited E-cadherin levels, hence reducing the rate of invasion and metastasis of lung cancer cells [32]. According to another study, the combination of high concentrations of curcumin and light attenuated the adhesion and attachment properties of A498, Caki1 and KTCTL-26 to HUVECs (human umbilical vein endothelial cells). According to the study’s results, the adhesive properties of Caki1 were completely blocked, while these properties were partially blocked in KTCL-26. Moreover, curcumin combined with light resulted in the complete downregulation of chemotaxis in Caki1, while upregulation of migration was observed in KTCTL-26 cells. Hence, a lower migration rate did not necessarily mean it was the outcome of low adhesion. The same study also portrayed the importance of evaluating cell surface proteins, such as integrins, due to their significant role in cell movement control. In all three cell lines, β1, β3, α3 and α5 were downregulated in the same manner by the synergistic effects of curcumin and light. β3 and α3 have been considered as prognostic markers in renal cancer because they are associated with a higher spreading capacity of the cells. β1 has the ability of promoting tumor growth as well as advancing it in metastasis, while α5 is expressed most abundantly on cell surface exerted paradoxical properties. However, the combination of light and curcumin was able to downregulate the expression of these integrins, which served as inhibitors of cell invasion and metastasis [33]. Later, Rutz et al. demonstrated that curcumin plus light radiations inhibit cell growth, proliferation, adhesion and metastasis of DU145 and PC3 prostate cancer cells. The potent anti-invasion activity was demonstrated by the dysregulation of integrin subtypes expression on DU145 and PC3 cells [34]. Another study strongly reported results of the expression of integrins and their effect on invasion and metastasis in bladder cancer. The results showed that the decreased motility of RT112, UMUC-3 and TCCSUP was due to suppression in the attachment of these cell lines to HUVECs. Moreover, the synergism of curcumin and light induced differences in integrin behaviors, especially in α3, thus inhibiting cell adhesion. Their results also revealed the role of α5 receptors in controlling both adhesion and chemotaxis in the three cell lines, whereas β1 solely acted on migration [35]. Another study published recently in 2022 investigated the role of curcumin in enhancing radiation therapy efficacy on glioblastoma cells in vitro. The results showed that curcumin, when combined with high linear energy transfer (LET) radiations, was able to significantly suppress glioblastoma cell invasion when compared to cells treated with curcumin alone or curcumin in combination with low γ-LET radiations [36].

### 2.2. The Hindrance of Metastatic/Proliferative/Invasive Pathways by Curcumin and Doxorubicin

Doxorubicin (DOX) is derived from the fungus *Streptomyces peucetius* var. *caesius* and belongs to the antibiotic family of anthracyclines. Doxorubicin mainly targets DNA molecules by intercalating between their base pairs, which ultimately leads to inhibition of topoisomerase II. The DOX-mediated stabilization of topoisomerase II halts the replication process by inhibiting DNA resealing [37]. Currently, DOX is the most efficient chemotherapeutic agent for the treatment of breast cancer and has portrayed an approximate response value of 35% in metastatic breast cancer [38]. However, DOX has exhibited some life-threatening effects, which hindered its clinical use where the appearance of cardiotoxicity has been the most critical side effect. Additionally, cancer cells in some patients might develop DOX resistance through the modifications of various pathways, eventually promoting continuous growth and survival despite the chemotherapy. As such, using DOX for long-term treatment might be challenging [39].

A study has reported that high expression of Aurora A in MCF-7 cells is correlated with the promotion of tumorigenesis and with the decrease in sensitivity to chemotherapeutic drugs, such as doxorubicin. In fact, Aurora A is a regulatory kinase and an important regulator of proliferation, migration, invasion, metastasis and apoptosis, as detailed in a review published by Lin et al. [40]. After treating MCF-7 cells with CUR + DOX, Aurora A was downregulated in a time-dependent manner and showcased a drastic reduction in the proliferation rate. Nevertheless, p21 levels were also detected to observe an analogy between Aurora A inhibition and the protein itself. The synergistic effects of curcumin and doxorubicin were also able to inhibit the expressions of p53; however, the complete mechanism was not clarified [41]. Another study investigated the enhanced anti-cancer activity of doxorubicin when combined with curcumin on the gastric adenocarcinoma cell line (AGS) in vitro. The results revealed more prominent anti-proliferative, pro-apoptotic, anti-invasive and anti-metastatic activity of doxorubicin when co-administered with curcumin on AGS cells. In this study, DOX + CUR had a significant higher effect on cell viability when compared to doxorubicin or curcumin alone. Similarly, alterations in cell morphology, such as membrane damage, reduced cell size and cell shrinkage, were significantly higher in AGS treated with DOX + CUR than doxorubicin or curcumin monotherapy. This study further mentioned the enhanced activity of doxorubicin when combined with curcumin on cell motility. Their results showed a dose-dependent inhibition of invasion and migration as revealed by the scratch wound-healing assay and the transwell migration assay performed on AGS cells in vitro [42].

From a different perspective, a study by Zhou et al. aimed at developing new strategies by creating multi-pH-sensitive polymer-drug conjugates mixed with micelles for efficient delivery of doxorubicin and curcumin. This strategy was performed on MDA-MB-231 breast cancer cells to investigate the efficient co-delivery of drugs and the suppression of tumor metastasis in breast cancer cells. First, the polymeric micelles showed a synergistic anti-proliferative effect on MDA-MB-231 cells in a dose-dependent manner. Moreover, a significant inhibition of tumor cell invasion was observed upon treatment with DOX + CUR as compared to doxorubicin and curcumin alone. An important process promoting tumor metastasis is transendothelial migration (TEM). The results showed a significant inhibitory effect of DOX + CUR on the migration of MDA cells across the HUVECs -coated wells. In vivo studies noted the superior role of doxorubicin when combined with curcumin on the suppression of proliferation and pulmonary metastasis of cancer cells [43]. A summary of the molecular targets of DOX + CUR identified to date is depicted in Figure 3.

### 2.3. The Hindrance of Metastatic/Proliferative/Invasive Pathways by Curcumin and 5-Fluorouracil

Pyrimidine analogue of uracil 5-fluorouracil (5FU) has a carbon at position 5 instead of a hydrogen atom. This agent exhibits multiple sources of therapies due to its anti-metabolite and anti-cancer properties. In fact, once 5FU enters the cells, it is directly converted to several active metabolites. These active metabolites suppress proliferation by interfering with DNA synthesis through the inhibition of thymidine formation [27]. Despite its various clinical uses, 5FU has exhibited long-term side effects on the cognitive side as well as nausea, cardiotoxicity and hepatotoxicity, which restricted its usage in breast cancer patients. Another limitation to 5FU is its resistance developed by cells, which mitigates its clinical use against different types of cancers.

Studies have shown that the combination of curcumin with 5-fluorouracil reduces NNMT-related resistance and downregulation via the suppression of p-STAT3. Firstly, by measuring the IC_50_ of low-dosed CUR + 5FU, the drastic reduction in the values in the two cell lines SW480 and HT-29 showed decreased proliferation and resistance to 5FU. This synergy also downregulated the mRNA expression of NNMT. Even though the authors reported a decrease in p-STAT3 in the SW480 cell, one cannot conclude that NNMT inhibition is due to low p-STAT3 expression [44]. To further investigate the synergistic effects of CUR and 5FU, hepatocellular carcinoma cell lines, as well as mice, were used. The increasing dose of CUR and constant concentration 5FU on SMMC-7721 cells caused an increase and then a decrease in nuclear NF-Kb, which means that the synergism of both agents inhibited the transfer of NF-kB from the cytoplasm to the nucleus (Figure 4). Moreover, COX-2 protein was downregulated in SMMC-7721, Bel-7402, HepG-2, MHCC97H and L02 [45]. Curcumin was further investigated for its potent role in reducing CAF (cancer-associated fibroblast)-induced resistance to 5-FU in tumor cells through the suppression of the JAK/STAT3 signaling pathway [46]. Another study revealed that increasing the concentration of curcumin on HCT-116 cells that are resistant to 5FU increased sensitization and showed decreased rates in proliferation as well as increased expression of TET1, NKD2 and Vimentin, whereas a downregulation of β-catenin, E-cadherin, TCF4 and Axin expression was reported. This study also mentioned that TET-1 was responsible in the inhibition of covalent bonding between cytosine and methyl catalyzed by methyltransferase, which partakes in demethylation. TET-1 was deemed as a novel cancer suppressor gene: the increased levels in this study showed that it upregulated NKD2, which ultimately inhibited the WNT pathway. Nevertheless, Pax-6 acted as a transcriptional mediator in TET-1 expression, where NKD2 and TET-1 were also increased upon its upregulation [47] (Figure 4).

### 2.4. The Hindrance of Metastatic/Proliferative/Invasive Pathways by Curcumin and Paclitaxel

Paclitaxel (C_47_H_51_NO_14_) is a well-known anti-cancer drug that is produced in the bark and needles of *Taxus brevifolid*. Paclitaxel has anti-tumor effects, mainly leading to mitotic arrest [48]. Paclitaxel promotes microtubule stabilization, thus preventing cell cycle progression, mitosis and growth of several types of cancers [49]. There is an increasing number of studies that have revealed that curcumin, in combination with paclitaxel, promotes inhibition of cell migration and thus inhibition of mitosis in various cell lines (Figure 5).

One of the many studies on that combination investigated its effects on ovarian cancer, both in vitro and in vivo. There was significant inhibition of cell migration in the SKOV3 cell line in response to treatment with paclitaxel only, as well as combined with curcumin. However, paclitaxel alone was not enough to affect the migration of the multi-drug-resistant cells of that same line. When combined with curcumin, however, the susceptibility of the multi-drug-resistant cell line to the treatment was restored, and a dose-dependent inhibition of metastasis was observed [50]. Even though the authors did not further investigate the molecular pathway underlying the inhibition of migration, previous studies in the literature provide a possible explanation: paclitaxel promoted the activation of NF-κB in breast cancer cells while curcumin inhibited its expression by inhibiting IκBα kinase activation. Furthermore, they revealed that this combination suppressed metastatic proteins, such as VEGF, MMP-9 and intercellular adhesion molecule-1, thus leading to the suppression of metastasis [51] (Figure 5).

Recently, new research has investigated molecular pathways underlying the anti-metastatic potential of curcumin combined with paclitaxel [52]. Vascular endothelial growth factor (VEGF), Cyclin D, and STAT3, all involved in metastasis, were shown to be effectively suppressed by curcumin treatment as well as in synergy with paclitaxel. The results reveal a downregulation in the gene expression of the three factors, with an upregulation of the pro-apoptotic caspase 9 (Figure 5). Overall, a potent regulation of metastasis was observed when cancer cells were exposed to curcumin alone or in combination with paclitaxel as compared to the effect of the chemotherapeutic drug alone [52].

### 2.5. The Hindrance of Metastatic/Proliferative/Invasive Pathways by Curcumin and Berberine

Berberine is an isoquinoline alkaloid isolated from the Chinese herb *Coptis chinensis*. It also has anti-proliferative activity reported in gastric cancer cells and anti-metastatic effects reported in breast cancer cells when combined with chemotherapy.

A combination of curcumin, berberine and quercetin was shown to effectively inhibit the expression of E-cadherin, mesenchymal N-cadherin, β-catenin, CD44 marker and MMP9 in triple-negative breast cancer cells (Figure 5). This cancer type is characterized by a lack of expression of estrogen, progesterone and human epithelial growth factor (HER2) receptors, making it difficult to treat by hormonal therapy and rendering it better managed by traditional cytotoxic chemotherapy. In the context of breast cancer, both E- and N-cadherins are involved in the EMT process and promote oncogenesis and metastasis. Similarly, CD44 surface adhesion receptor facilitates invasion of cancer cells and their migration. Thus, inhibition of these markers by the multi-compound combination has anti-proliferative and significant anti-migratory consequences that potentiate each compound’s individual effect [53].

Another study aimed to evaluate the anti-cancer properties of curcumin, berberine and 5-Fu, alone or in combination, on breast cancer cells. The strongest effect on cell growth and invasion of MCF-7 cells was noticed when the cells were treated with CUR + BER + 5-Fu compared to a control or to each drug alone. Their study demonstrated the potential anti-apoptotic and anti-invasive properties of curcumin in combination with berberine and 5-Fu [54].

### 2.6. The Hindrance of Metastatic/Proliferative/Invasive Pathways by Curcumin and Docetaxel

Docetaxel (DTX) is sold under the brand Taxotere is part of the taxane family, a class of diterpenes that originate from plants of the genus Taxus. It is a chemotherapeutic agent typically useful in many cancer types, especially unresectable and metastatic cases. It interferes with microtubules assembly by binding β-tubulin, thus leading to cell cycles arrest at the G2/M phase. Docetaxel also downregulates the expression of the anti-apoptotic, pro-proliferative Bcl-2 protein.

A treatment of docetaxel combined with curcumin demonstrated great efficiency, both in vitro and in vivo, in pancreatic cancer, lung cancer, glioma (brain tumor) and esophageal squamous cell carcinoma [55]. The molecular pathways involved are illustrated in Figure 6. Particularly, in PANC-1 pancreatic cancer cells, the combination led to the downregulation of MMP2 and MMP9, both pro-invasive and pro-metastatic metalloproteinases. This is mediated through an upregulation of tissue inhibitor matrix metalloproteinase 1 (TIMP-1), a natural inhibitor of MMPs. The same combination strategy was used to treat esophageal squamous cell carcinoma using ESCC KYSE150 and KYSE510 cells, demonstrating its ability to weaken the healing capacity of the cancer cells and inhibit their invasion [56].

Another study demonstrated a different intervention on the usage of DTX and CUR against pancreatic cancer. GE11-DTX-CUR NPs are nanoparticles synthesized to deliver DTX and CUR into tumor cells. The NPs have GE11 target EFGR peptide on the surface of these cells. LNCaP cells were used in this study and were subject to in vitro and in vivo assays. A low IC_50_ was observed, which is the minimal concentration of a drug required to achieve 50% inhibition of cell proliferation. In addition, CUR inhibits the PIK3/AKT pathway involved in the proliferation, apoptosis and metastasis of the tumor. CUR also lowers the endoplasmic reticulum stress, which is a pathway that allows tumor survival and expansion. Therefore, CUR and DTX introduced as GE11-DTX-CUR NPs constitute a synergistic antitumor treatment for pancreatic cancer [57]. Curcumin and docetaxel co-loaded poly lactide-co-glycolide (PLGA) nanoparticles were used to target U87 glioma cells and bEND.3 endothelial cells [58]. These nanoparticles can surpass the blood–brain barrier and deliver the drug (DTX + CUR). A low IC_50_ was obtained, indicating the efficiency of the drug against invasion and metastasis.

### 2.7. The Hindrance of Metastatic/Proliferative/Invasive Pathways by Curcumin and Metformin

Metformin was FDA-approved in 1994 and is best known as an antidiabetic agent used in type 2 diabetes mellitus patients. It can further be used in combination with other agents, such as curcumin, against cancer. Metformin is known as a suppressor of mTOR activity by activating ataxia telangiectasia mutated (ATM) and liver kinase B1(LKB1) as well as adenosine-monophosphate-activated kinase (AMPK), thus preventing protein synthesis and cell growth.

Several studies were conducted to support that metformin in combination with curcumin could be a potential drug against different types of cancer. HCC cell lines HepG2 and PLC/PRF/5, involved in hepatocellular carcinoma, were treated with metformin and curcumin. An increase in the inhibition rate of invasion for HepG2 and PCL/PRF/5 and the downregulation of MMP2 and MMP9 in HepG2 cells resulted from the co-administration of curcumin and metformin. Moreover, an upregulation in the expression of tumor suppressors PTEN and p53 was observed as well as an inhibition of NF-kB levels upon treatment [59]. Therefore, a combination of metformin with curcumin could be used to affect invasion and metastasis by inhibiting MMP2 and MMP9 expression and the PTEN/PI3K/Akt/mTOR/NF-kB signaling pathway (Figure 6).

A study on gastric adenocarcinoma was also able to support the idea of this combination therapy against cell proliferation. The treatment consisting of metformin plus curcumin played a key role by reducing cell migration and invasion in a dose- and time-dependent manner, thus affecting the metastatic potential of human AGS gastric cells [60].

### 2.8. The Hindrance of Metastatic/Proliferative/Invasive Pathways by Curcumin and Gemcitabine

Gemcitabine (GEM) (a pyrimidine nucleoside antimetabolite) is a cytotoxic agent showing promising results when used in treatment against solid tumors. Since gemcitabine is a pyrimidine nucleoside analogue, it can be incorporated into DNA, thus interfering and blocking DNA synthesis [61]. It has been proved that gemcitabine in combination with curcumin has a synergistic activity against cancer with low toxicity.

A study performed on pancreatic cancer cells evaluated the effect of GEM in combination with curcumin. The results showed a more efficient decrease in the number of invasive pancreatic cancerous cells when exposed to GEM plus curcumin [55]. In fact, this combinatorial treatment was found to exhibit higher anti-proliferative and pro-apoptotic activity on PANC-1 cells in vitro. Further, they performed wound healing assays to determine the inhibitory effect of CUR + GEM on migration and their findings were supported by the significant reduction in N-cadhinerin and Vimentin expression. Further, they investigated the anti-invasion properties of curcumin in combination with GEM in PANC-1 cells. Their results showed a significant decrease in the rate of invasive cells along with an upregulation in the expression of TIMP1 in a dose-dependent manner. Moreover, a downregulation in the expression of MMP2 and MMP9 was noticed upon treatment with CUR + GEM. Overall, this study demonstrates the synergistic anti-proliferative, pro-apoptotic, anti-invasive and anti-migration effects of GEM when combined with curcumin (Figure 6).

In addition, a positive correlation between GEM plus curcumin treatment and the preservation of quality of life in patients was displayed. In metastasized cancer, high baseline levels of IL-6 and sCD40L are encountered. IL-6, an immunosuppressive cytokine, exhibits a positive correlation with sCD40L (activate T lymphocytes), thus allowing tumor growth. However, no increase in these biomarkers took place with the intake of this treatment [62]. 

Another study involving pancreatic cancer and treatment with GEM plus curcumin was conducted on HPAF-II and PANC-1 cell lines [63]. In this study, they used a supermagnetic iron oxide nanoparticle-curcumin combined with gemcitabine (SP-CUR + GEM) to enhance drug delivery in pancreatic cancer cells. Their combinatorial treatment showed an increase in the expression of E-cadherin, which is involved in metastasis inhibition. An important pathway, the Sonic hedgehog (SHH) pathway, involved in the regulation of cell progression, was further studied. Changes in key regulatory proteins of SHH were determined, including Gli-1 and Gli-2, upon treatment with curcumin and gemcitabine (Figure 6). Moreover, the SP-CUR enhanced the uptake of GEM into the cells through the inhibition of the CXCL-12, CXCR-4 pathway, highly involved in the regulation of growth, survival and metastasis. Another study performed on GEM-resistant lung cancer cells demonstrated the role of curcumin in improving the sensitivity of A549 cells to gemcitabine. The combination treatment inhibited invasion and migration of lung cancer cells, as shown by the downregulation of MMP9, Vimentin and N-cadherin, while overexpressing the E-cadherin expression level [64] (Figure 6).

### 2.9. The Hindrance of Metastatic/Proliferative/Invasive Pathways by Curcumin and Carboplatin

Carboplatin, sold under the trade name Paraplatin, is a chemotherapy medication used to treat several types of cancer. However, the use of carboplatin was limited due to its side effects, such as myelosuppression. Interestingly, a previous study revealed the potent role of curcumin in reversing carboplatin-induced toxicity and myelosuppression [65].

The combination of curcumin with carboplatin showed prominent results on the inhibition of tumor cell proliferation, invasion and migration in lung and breast cancer cells. A study conducted by Kang et al. investigated the synergistic anti-proliferative and anti-metastasis properties of curcumin and carboplatin in lung cancer cells [66]. Based on the results obtained, the combination treatment revealed a synergistic inhibition of cell proliferation in a dose-dependent manner in A549 lung cancer cells along with changes in cell morphology, including cell shrinkage, loss of cell-to-cell contact, membrane blebbing and partial detachment. They further investigated the rate of migration and invasion via wound healing and Matrigel cell invasion assays. Their data suggest a significant inhibitory effect of migration and invasion upon co-treatment with curcumin and carboplatin. To further understand the anti-invasive activities of this combination, they assessed two different metastasis regulators, namely MMP2 and MMP9. They found that the combination of curcumin and carboplatin significantly suppresses metastasis through the inhibition of these two markers. 

Another study on breast cancer revealed that this combinatorial treatment inhibits cell proliferation and promotes apoptosis in CAL-51 and MDA-MB-231 cells in vitro. They evaluated the effect of curcumin when combined with carboplatin on colony formation and their results showed a dose-dependent inhibition of colony formation in both cell lines. At a molecular level, they reported a downregulation of Rad51 expression, a DNA repair gene and an upregulation of γH2AX expression, involved in DNA damaging [67]. It was previously reported that Rad51 protein is overexpressed in cancer cells. Dysregulation or inhibition of this protein was proposed to be responsible for the suppression of metastasis and invasion [68]. Taken together, the study by Wang et al. suggests a potent anti-metastasis activity of curcumin and carboplatin through the inhibition of Rad51 in breast cancer cells.

### 2.10. The Hindrance of Metastatic/Proliferative/Invasive Pathways by Curcumin and Other Drugs

Few recent studies were identified in the past five years that investigate the superior role of curcumin when combined with other drugs in the inhibition of tumor cell proliferation, invasion and metastasis; these are summarized in Table 1.

A study published recently evaluated the potential therapeutic effects of curcumin and resveratrol hybrids on colorectal cancer cells, namely SW620 and SW480. Their results showed that different hybrid molecules induce apoptosis through the caspase-dependent pathway in both cell lines. Further, they demonstrated a direct interaction between the hybrids and MMP7 molecules, suggesting the modulation of MMP7 catalytic activity, hence preventing cancer cell progression [69]. Similarly, Panda et al. investigated several hybrids of curcumin and dichloroacetate (DCA), a pyruvate dehydrogenase kinase 1 (PDK1) inhibitor, able to promote apoptosis in breast cancer cells. In vitro and in vivo results revealed the anti-proliferative and anti-metastatic activity of these conjugates in breast cancer [70]. Their data suggest that curcumin and DCA hybrids significantly reduced cell viability and colony formation in MDA-MB-231 and T47D cells in a dose-dependent manner.

Moreover, the combination of two phytochemicals, curcumin and luteolin, showed promising anti-cancer activity on colon cancer [71]; curcumin and luteolin synergistically inhibit proliferation and migration of colon cancer cells, as shown by the wound healing assay. Moreover, protein expression analysis revealed the suppression of Notch-1 and TGF-β in vitro and in xenograft mice. These data suggest that curcumin and luteolin are effective in suppressing colon cancer cell proliferation and metastasis via the Notch1 and TGF-β signaling pathway. A different study evaluated the anti-cancer activity of curcumin when combined with aprepitant, a drug known for its antitumor properties on various cancers [72]. A study conducted on hepatocellular carcinoma demonstrated that liposome conjugates of curcumin and aprepitant drug are able to reduce ECM deposition through the inhibition of collagen synthesis. Wound healing assays showed a higher inhibition of the migration rate of SMMC-7721 cells when treated with liposome conjugates of curcumin and aprepitant. Their findings were further confirmed by the transwell migration assay. Their results revealed that the combination of curcumin with aprepitant is able to suppress tumor cell growth and migration as well as inhibiting lung metastasis in vivo [73]. The combination of curcumin was further studied with two other bioactive compounds, namely thymoquinone (TQ) and 3,3′-diindolymethane (DIM), on A549 lung cancer cells and HepG2 liver cancer cells in vitro [74]. The authors assessed the anti-metastasis activity of curcumin when combined with TQ and DIM, alone or together, in A549 cells by performing wound healing and colony formation assays. Their results revealed a significantly higher inhibitory effect of migration as well as colony formation in a dose-dependent manner as compared to control samples. Furthermore, this combination upregulates caspase-3 protein levels while it significantly downregulates PI3K and AKT levels in A549 lung cancer cells. Their data show that the combination therapy suppresses tumor cell proliferation as well as migration activity via the inhibition of the PI3K/Akt pathway in A549 and HepG2 cells. Additionally, Shao et al. assessed the synergistic anti-proliferative and anti-metastatic activity of curcumin when combined with a new biflavonoid, wikstroflavone B (WFB), on nasopharyngeal carcinoma cells (NCC), namely CNE-1 cells, in vitro [75]. Migration assays revealed a significantly higher inhibition of CNE1 tumor cell migration as compared to the control group. Curcumin along with WFB inhibits tumor cell proliferation in a dose-dependent manner, as revealed by the dysregulation of several tumor growth markers, including survivin, cyclin D1, p53 and p21 gene expression. Further, their data revealed the modulation of tumor invasion and metastasis markers, such as MMP-2 and MMP-9, as well as FAK gene expression in CNE1 cells. Since the qRT-PCR data suggest that FAK was one of the most highly regulated genes upon treatment with CUR and WFB, they aimed at investigating the regulation of the FAK/STAT3 pathway in CNE1 cells. Western blot analysis showed a significant decrease in the protein expression level of p-FAK and p-STAT3 in CNE1 cells upon CUR + WFB treatment. To further confirm that the FAK/STAT3 pathway is involved in the previously determined anti-cancer activity, pretreatment of CNE1 cells with FAK inhibitor was performed. Their results showed a more potent inhibitory effect of cell proliferation and migration when the NCC were pre-exposed to the FAK inhibitor as compared to CUR + WFB treatment. The data reported in this study suggest that the combination of curcumin and WFB plays a crucial role in the regulation of NCC growth, proliferation, invasion and metastasis in a FAK/STAT3 dependent manner.

## 3. Discussion

Combination therapy is an emerging approach effective in increasing the treatment efficacy of cancer by limiting the major drawbacks of various chemotherapeutic drugs. Many studies in the literature target the combination of curcumin with various therapeutic agents, revealing a synergistic inhibitory effect of several pathways involved in the regulation of cell proliferation, metastasis and invasion. The data presented in this review outline the underlying molecular mechanisms of curcumin with various drugs; the results obtained provide solid evidence that curcumin combination chemotherapy interferes with multiple distinct cellular factors, as depicted in Appendix A. In brief, this combinatorial treatment was found effective in the suppression of migration proteins, including MMPs, VEGF and cytokines, as well as several signaling pathways, such as NF-κB and JAK/STAT pathways.

However, the clinical therapeutic application of these combinatorial regimens has been hampered by major limitations: the hydrophobicity of curcumin results in poor bioavailability due to its low absorption in the plasma, non-uniform bio-distribution and poor localization in the targeted cancerous tissues, hence increasing the drug toxicity [76]. For this reason, a new perspective using curcumin along with polymer-based nanocarriers is currently under investigation. These conjugates provide effective drug delivery, reduce drug toxicity and increase drug stability [77]. Scientists are able to control several factors, including the shape, size and composition of the nanoparticles, to ensure effective co-delivery of selected chemotherapeutic drugs into the tumor microenvironment [78]. Recently, these nanocarriers are being thoroughly investigated in combination therapy with several pharmaceutical agents that were presented in the results section to pave the way for promising clinical applications. Multi-pH-sensitive conjugate micelles of curcumin and doxorubicin synergistically inhibited cell proliferation and invasion of breast cancer cells [43]. Similarly, nanoparticle formulation of encapsulated paclitaxel and curcumin was able to inhibit tumorigenesis of ovarian cancer cells due to the efficient co-delivery of these drugs [50]. The various nanocarrier formulations, including polymeric nanoparticles, micelles, nanoliposomes, polymer-drug conjugates, dendrimers, hydrogels, nanocapsules and exosomes, are all promising strategies for the use of curcumin in combination with different anti-cancer agents [77].

It is noteworthy that, despite the recent advancements in polymeric nanoparticles therapy, scientists are still facing several challenges, such as the high cost of the nanoformulations as well as their long-term toxicity, which requires further investigation. However, all the current studies in this field show promising results and pave the way for in-depth clinical studies of curcumin nanoformulation in synergy with chemotherapeutic drugs to inhibit tumor metastasis and invasion.

## 4. Methods

### 4.1. Search Strategy

The PRISMA guidelines were followed to evaluate the literature of curcumin and its synergistic effects with multiple chemotherapeutic drugs. MEDLINE/PubMed and Scopus databases were referred to up to 2022. The reason behind choosing these two databases was due to their easy navigation and access to our scope of review in cancer chemoprevention. The search keywords applied were: (curcumin) AND (metastasis OR metastatic pathways) AND (proliferation OR proliferative pathways) AND (invasion OR invasive pathways) AND (combination therapy OR in combination with chemotherapeutic agents) AND (synergism OR synergistic effects OR in synergy) AND (inhibition OR inhibiting OR prevention). In the course of article searching for our literature, filters were applied to select for studies published within the last five years, while older studies were also chosen to be referenced in the introduction. 

### 4.2. Selection of Studies

#### 4.2.1. Inclusion Criteria

In search of the data for the systematic review, the inclusion criteria attributed to it included: (i) mechanisms of suppressing the different pathways and proteins involved in metastasis, proliferation and invasion; (ii) published articles were of pure research; (iii) published articles were written in English and free access to their full texts; (iv) in vitro human cancerous cell lines as well as in vivo animal studies; (v) combination therapy of curcumin along with various chemotherapeutic drugs in inhibiting metastasis, proliferation and invasion; (vi) studies that solely used curcumin as a means of treating multiple cancerous cell lines in different pathways.

#### 4.2.2. Exclusion Criteria

On the other hand, the exclusion criteria encompassed were: (i) studies that used the different therapeutic effects of curcumin, such as its anti-inflammatory role or its role in ameliorating cardiovascular pathologies; (ii) studies that used the combination of curcumin and chemotherapeutic drugs in regulating other events in cancer, such as apoptosis, necrosis or cell cycle death; (iii) studies that used only curcumin as a therapeutic agent in inhibiting metastasis, proliferation and invasion; (iv) published articles that are reviews, letters to editors or authors and commentaries.

#### 4.2.3. Data Extraction

The data extracted for the included studies were the type of pharmacological intervention, the methods, the cell lines, the molecular outcomes and study conclusions.

## 5. Conclusions

The current review is the first review to assess the various combination therapies between curcumin and multiple drugs that have portrayed inhibitory properties against metastatic pathways. The synergistic effects of curcumin in combination with synthesized chemical pharmaceuticals showed potential treatment in the suppression of different mechanisms and proteins involved in cell invasion, proliferation and metastasis, including SIRT, JAK/STAT, MAPK, P13K/Akt, Wnt/β-catenin, Notch and NF-κB as well as uPA, MMPs, VEGF and interleukins.

However, the current studies are limited to a few aspects in the sense that there is a lack of evidence regarding the long-term toxic properties of such synergy as well as determining the core proteins that partake in the inhibition process of the pathways. Further research is required to comprehend the complete complex regulatory networks that contribute to the anti-cancer actions of curcumin in combination chemotherapy. Perhaps future research can expand on regulatory proteins in different metastatic pathways. Since the literature shows that the most significant anti-cancer properties are observed upon combination of curcumin with other drugs, it is highly possible to consider integrating curcumin in clinical treatment, particularly with the implementation of new promising nanoformulations that increase its bioavailability. Such treatments are undoubtedly needed to determine the safety, tolerability and effectiveness of curcumin in combination with existing anti-cancer drugs.

## Figures and Tables

**Figure 1 plants-11-02137-f001:**
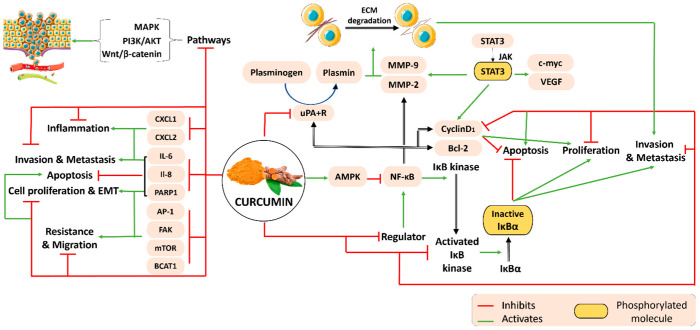
Molecular pathways underlying the anti-tumor properties of curcumin on various types of cancer.

**Figure 2 plants-11-02137-f002:**
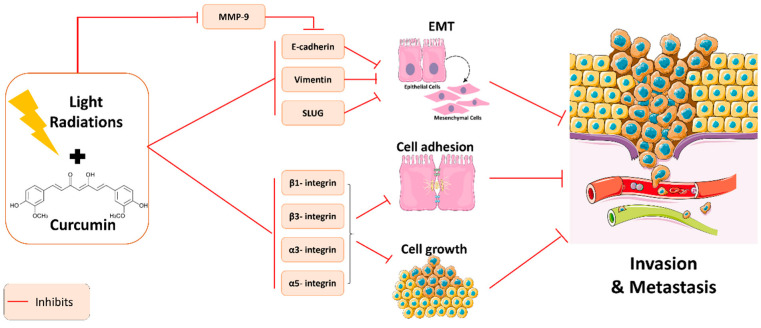
Molecular pathways underlying the anti-metastatic and anti-invasive properties of curcumin in combination with light radiation therapy.

**Figure 3 plants-11-02137-f003:**
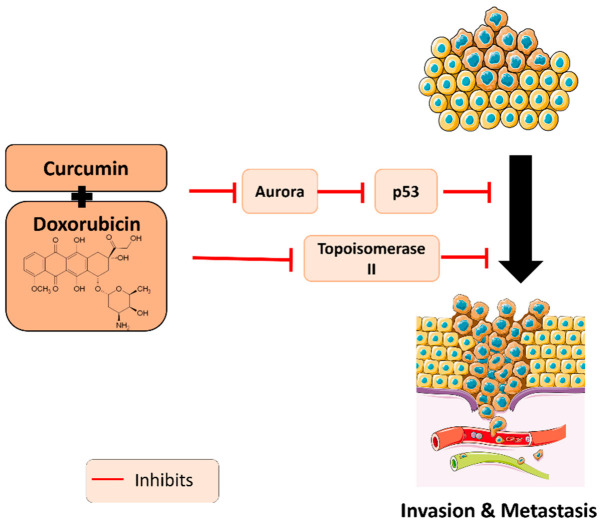
Molecular pathways underlying the anti-metastatic and anti-invasive properties of curcumin in combination with doxorubicin.

**Figure 4 plants-11-02137-f004:**
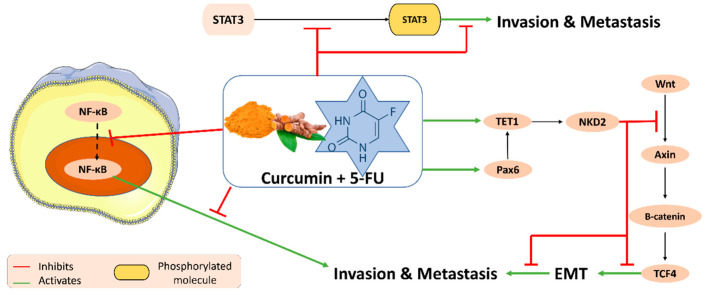
Molecular pathways underlying the anti-metastatic and anti-invasive properties of curcumin in combination with 5-fluorouracil (5-FU).

**Figure 5 plants-11-02137-f005:**
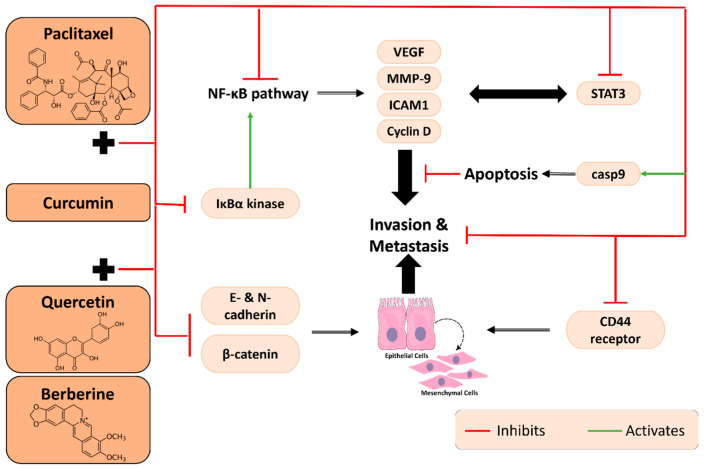
Molecular pathways underlying the anti-metastatic and anti-invasive properties of curcumin in combination with paclitaxel or with quercetin and berberine.

**Figure 6 plants-11-02137-f006:**
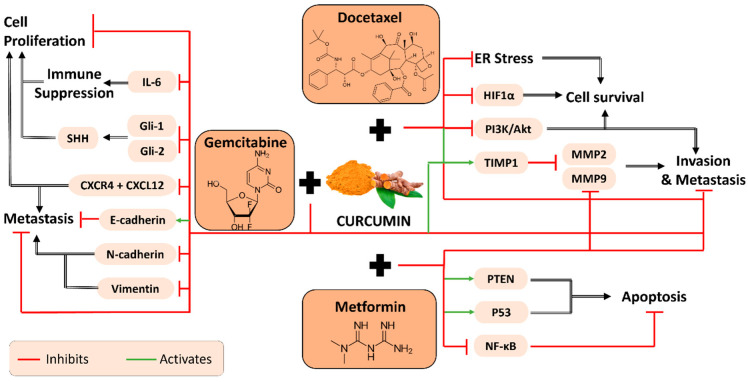
Molecular pathways underlying the anti-metastatic and anti-invasive properties of curcumin in combination with each of the three chemotherapeutic drugs: docetaxel, metformin and gemcitabine.

**Table 1 plants-11-02137-t001:** Table summarizing the combination therapy in the past five years of curcumin and chemotherapeutic agents resveratrol, DCA, luteolin, aprepitant, WFB, thymoquinone and DIM in metastatic, invasive and proliferative cancer therapy.

Reference	Cancer Type (Cell Lines)	Type of Intervention	Methods	Molecular Outcome	Study Conclusion
[69]	Colorectal cancer cells	Pharmacological: curcumin + resveratrol	In vitro: staining for mitochondrial membrane potential and plasma membrane integrity Molecular docking	Promotes apoptosis by inducing mitochondrial instability in both cell lines Promotes apoptosis via caspase-dependent pathway P53-independent cell death activity of curcumin + resveratrol G0/G1 cell cycle arrest upon treatment with both drugs Alteration in MMP7 protein upon combination therapy	Combination of curcumin and resveratrol has synergistic antiproliferative, antimetastatic and pro-apoptotic effects in colorectal cancer cells
[70]	Breast cancer cells	Curcumin + dichloroacetate	In vitro MTT assay, colony formation assay, molecular docking	Antiproliferative activity of combination therapy on cancer cells Curcumin and DCA inhibits colony formation in both cell lines	Curcumin and dichloroacetate inhibit breast cancer cells’ survival, proliferation and metastasis
[71]	Colon cancer cells	Curcumin + luteolin	In vitro: cell proliferation assay, wound-healing assay, Western blot	Synergistic effect of curcumin and luteolin on colon cancer cells Anti-proliferative effect on both cell lines Inhibition of tumor metastasis revealed by wound-healing assay Suppression of Notch-1 and TGF-β	Curcumin and luteolin synergistically inhibit the proliferation as well as invasion and metastasis of colon cancer cells via the suppression of the Notch-1 and TGF-β signaling pathway
[73]	Hepatocellular carcinoma	Curcumin + aprepitant	In vitro: cytotoxicity assay, cell migration assay In vivo: lung metastasis	Efficient delivery of both drugs via the liposome conjugates Anti-proliferative effect Inhibition of metastasis revealed by wound healing assay Reduced ECM deposition via the inhibition of collagen IV production Inhibition of lung metastasis when mice were treated with both drugs as compared to control groups	Co-delivery of curcumin and aprepitant via modified liposome conjugates, effective in suppressing tumor cell proliferation, invasion and metastasis as well as inhibiting lung metastasis in vivo
[74]	Lung and liver cancer cells	Curcumin + thymoquinone + 3,3′-diindolymethane (DIM)	In vitro: cell proliferation assay, migration assay, colony formation assay and Western blot analysis	Antiproliferative effect of combinatorial treatment on lung and liver cancer cells Inhibition of A549 and HepG2 cell migration Decreased expression of PI3K and AKT proteins level	Combination of curcumin with thymoquinone and DIM exhibits anti-proliferative as well as anti-metastatic activity on lung and liver cancer cells
[75]	Nasopharyngeal carcinoma cells (NCC)	Curcumin + wikstroflavone B (WFB)	In vitro: cytotoxicity assay, colony formation assay, wound healing assay and transwell migration assay	Anti-proliferative activity on NCC Dysregulation of cyclin D1, survivin, p53 and p21 Anti-metastatic activity on NCC Downregulation of MMP-2 and MMP-9 Inhibition of the FAK/STAT3 signaling pathway	Curcumin when combined with WFB inhibits nasopharyngeal carcinoma cell proliferation, invasion and metastasis via the suppression of the FAK/STAT3 signaling pathway

## Data Availability

Not applicable.

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
