# Peer review of "The Synergistic Effects of Curcumin and Chemotherapeutic Drugs in Inhibiting Metastatic, Invasive and Proliferative Pathways"

_plants, 2022, doi:10.3390/plants11162137_

Round 1
Reviewer 1 Report
The Authors performed a literature review on Curcumin entitled "The Synergistic Effects of Curcumin and Chemotherapeutic
Drugs in Inhibiting Metastatic, Invasive and Proliferative Pathways"
The manuscript is interesting, and well written, however before the editor makes a decision on the work, I recommend minor revisions be carried out,
In my point of view authors should improve the abstract, the first part is good, however, authors should show the main advances that their work brings and should show the main findings and not focus only on the method used to search the articles used in the review.
The introduction citations should be at the end "and ovarian cancer [7]" correct for and ovarian cancer [7]" there are other similar errors throughout the manuscript, please review.
The results are well presented, but the discussion does not seem to have been discussed in depth, I recommend that instead of separating, leaving a topic of Results and discussions together, it is better to understand the manuscript.
I don't agree with how the conclusion was made, to me, it looks like methods. I would have liked to have read what the authors found that was new by bringing together all the information and literature, advances, and perspectives.
Reviewer 2 Report
The current review discusses the synergistic effects of curcumin combined with multiple drugs and radiotherapy in the inhibition of various types of cancer cells proliferations, invasion and metastasis from a molecular point of view.
Comments:
Keywords: it would be better to replace herbal medicine with phytotherapy
Introduction:
1.Line 37: curcumin exerts cytotoxicity on cancer cell lines not anti-cancer effects. Please replace.
2. line 53: you forgot to write what Fig 1 represents. Also, Figure 1 is not fully explained. Please complete.3. lines 92, 94, 99, 110 - you wrote: other drugs, several drugs, nany chemotherapeutic drug, such drugs. Please give some examples of such drugs. You wrote only about Cisplatin.
Results:
1. pgs7-8, lines 298-300, you wrote: a better control of metastasis was observed in cells treated by curcumin on its own as compared to those treated with paclitaxel only”. This statement is strange.
2. pg 8, lines 306-7, delete please: “Different nutraceutical properties, such as anti-inflammatory effects, have been reported 306 in the literature [53]”.
3. pg 8, lines 309-321 and Fig 5: this study includes quercetin and in my opinion is not suitable for the point 2.5 and for the present article.
4. pg 9, lines 345-46: you wrote about the combination docetaxel with octreotide modified curcumin. The modified curcumin it’s not the subject of the present review. So I consider you must delete this study.
5. pg 10, lines 392-93: please add the mechanism of action by which Gemcitabine exerts its action alone. Give some details concerning the fact that is a pyrimidine nucleoside antimetabolite.
6.It would be better to have a point 2.9 in which to write about Curcumin and Carboplatin. In this aim you need a little moore informations. Curcumin and other drugs will become 2.10
7. In the section 2.9 you don’t have any informations concerning curcumin and dichloroacetate.
8. It would be interesting to provide more information regarding the anticancer activity of curcumin with aprepitant, curcumin with Thymoquinone, curcumin with 3,3’-diindolymethane and curcumine with wikstroflavone B because are little known.
9. In my opinion Table 1 isn’t useful. Are presented the same results as in the subsections 2.1-2.8. You can leave Table 1 concerning Curcumin and other drugs (for subsection 2.9), for wich you gave less information in the corresponding subsection.
Conclusion: lines 559-60: Why do you refer to other diseases? Other diseases aren't the topic of the current review.

Reviewer 3 Report
In this review of "The Synergistic Effects of Curcumin and Chemotherapeutic Drugs in Inhibiting Metastatic, Invasive and Proliferative Pathways", the authors assesses the synergistic effects of curcumin with chemotherapeutic drugs and light radiation, in the regulation of various cancerous pathways, including the inhibition of metastasis, invasion, and proliferation. Further studies on long-term toxic properties as well as the biological inhibition process of strategy are still unexplored.
However, the following concerns should be addressed for further publishing consideration.
1. Please make an abbreviations summary at the end of this review, and it will help the readers easily understand.
2. Please illustrate the chemical structures of curcumin and the key drugs in Figures.
3. Please provide a subtitle for Figure 1 if possible.
4. In line 246, the IC50 should be subscripted for 50.
Round 2
Reviewer 3 Report
The authors have done all the necessary correction and now the manuscript could be accepted in this current version.
Author Response
The reviewer had no further comments. We wish to thank the academic editor for the thorough review of our manuscript and for the constructive feedback provided. The manuscript has been revised accordingly and the discussion has been re-written.